# Exploring Carob (*Ceratonia siliqua* L.): A Comprehensive Assessment of Its Characteristics, Ethnomedicinal Uses, Phytochemical Aspects, and Pharmacological Activities

**DOI:** 10.3390/plants12183303

**Published:** 2023-09-18

**Authors:** Widad Dahmani, Nabia Elaouni, Abdelhadi Abousalim, Zachée Louis Evariste Akissi, Abdelkhaleq Legssyer, Abderrahim Ziyyat, Sevser Sahpaz

**Affiliations:** 1Laboratory of Bioresources, Biotechnologies, Ethnopharmacology and Health, Department of Biology, Faculty of Sciences, University Mohammed First, Oujda 60000, Morocco; widad.dahmani@ump.ac.ma (W.D.); e.nabia@ump.ac.ma (N.E.); a.legssyer@ump.ac.ma (A.L.); ar.ziyyat@ump.ac.ma (A.Z.); 2Plant Tissue Culture Laboratory, Horticultural and Local Products Unit, Plant Production, Protection and Biotechnology Department, Hassan II Institute of Agronomy and Veterinary Medicine, 6202 Rabat-Instituts, Rabat 10112, Morocco; a.abousalim@iav.ac.ma; 3Joint Research Unit 1158 BioEcoAgro INRAE, University of Lille, University of Liège, UPJV, JUNIA, University of Artois, ULCO, 5900 Lille, France; zachee.akissi@univ-lille.fr

**Keywords:** carob, *Ceratonia siliqua*, phytochemistry, pharmacology, toxicity, traditional medicine

## Abstract

The carob tree (*Ceratonia siliqua* L.) is currently considered one of the most valuable fruit and forest trees in various fields and sectors of activity. It is a versatile plant, belonging to the Fabaceae family. It is widely used in traditional medicine to treat many diseases such as diabetes, hypertension, and gastrointestinal disorders, given that all its parts (leaves, flowers, pods, seeds, wood, bark, and roots) are useful and hold value in many areas. Its importance has increased significantly in recent years. Originating from the Middle East, it is recognized for its ecological and industrial significance. Previous studies conducted on *Ceratonia siliqua* L. have revealed the presence of several compounds, including polyphenols, flavonoids, carbohydrates, minerals, and proteins. The carob tree demonstrates antihypertensive, antidepressant, anti-obesity, and antihyperglycemic activities. This plant is known for its medicinal and therapeutic virtues. Moreover, it is particularly interesting to consider the pharmacological activities of the major phytochemical compounds present in the different extracts of this plant, such as phenolic acids, for example, coumaric and gallic acids, as well as flavonoids such as kaempferol and quercetin. Therefore, this review aims to analyze some aspects of this plant, especially the taxonomy, cytogeography, traditional uses, phytochemical constituents, and pharmacological activities of *Ceratonia siliqua* L., in addition to its biological properties.

## 1. Introduction

The carob tree (*C. siliqua* L.) is an evergreen perennial tree from the Fabaceae (Leguminosae) family. Originating in the Mediterranean region, it now populates many parts of the world, including North and South America, Africa, and Australia. The tree grows up to 15 m tall and boasts long, dark green leathery leaves [1,2]. Until 1980, the *Ceratonia* genus contained only one species, *C. siliqua*. However, another species, *C. oreothauma*, has since been discovered in East Africa and the Arabian Peninsula [3].

This plant has been utilized by humans since the ancient times. It is valued for its economic and culinary importance. Its seeds, also known as carob, are used as a food source for both humans and livestock [4,5]. High in carbohydrates and protein, they can be ground into a powder and used as a chocolate substitute [6]. The leaves, bark, and seeds have traditionally been used in medicine to treat various diseases, including diarrhea, diabetes, and hypertension [7,8]. In addition to its culinary uses, carob is believed to have several pharmacological activities, including antioxidant, antidiarrheal, antibacterial, antiulcer, and anti-inflammatory effects [9,10]. Rtibi et al., 2015, suggest using carob as a natural antioxidant in the form of a food supplement for the prevention of damage caused by oxidative stress [9]. In Morocco, the carob tree is particularly abundant in certain regions (Marrakech, El Ksiba, Khenifra, Beni Mellal, Meknes, Essaouira, Elhaouz, Kasba Tadla…); it provides farmers in these areas with a way to add value to land unsuitable for other crops. These regions could become the centers of carob production in Morocco. Nothing appears to hinder the success of the carob tree in these areas, with some specimens growing with minimal care and yielding interesting outputs [11].

This review aims to evaluate this species and its various aspects such as taxonomy, cytogeography, biological significance, and its therapeutic properties.

## 2. Materials and Methods

This review is the result of a comprehensive bibliographic analysis, aimed at amalgamating all available morphological, ethnomedicinal, ecological, phytochemical, pharmacological, and toxicological research on *C. siliqua* L. Our literature search utilized various scientific databases, including Scopus, ScienceDirect, Web of Science, Springer, JSTOR, PubMed, and the search engine Google Scholar. The primary keyword used in this study was the common name of *C. siliqua* L. (Carob), paired with the relevant research domain (e.g., ecology, pharmacology). The bibliography compiled comprises 166 references, ranging from original articles and reviews to theses, books, and book chapters, as well as one website, spanning publications from 1883 to 2023. We excluded unpublished theses and conference communications. All the references incorporated in this review were either in French or English, and we accessed them in their entirety or through informative abstracts. The chemical structures were cross-referenced using the open chemistry database PubChem and illustrated using ChemDraw Professional 15.0.

## 3. Systematic and Botanical Classification of the Carob Tree

The scientific name of the carob tree, *Ceratonia siliqua*, stems from the Greek “keras”, which means horn, and the Latin “siliqua”, referring to the hardness and shape of the pod [12]. Its common name originates from the Hebrew “kharuv” [2]. Its Arabic common name is “kharroub”, in English “carob” [13], in French “caroubier” [14], in Italian “sciuscella” [15], in Spanish “algarrobo” or “garrofer” [16], in Chinese “chang jiao dou” [2], in Portuguese “alfarrobeira”, and in German “karubenbaum” [17]. Carob is also known as St. John’s bread or locust bean [18]. The common name “carob” is sometimes incorrectly used to refer to other species, such as those of *Prosopis* sp. [19].

The species *C. siliqua* belongs to the genus *Ceratonia* L., the subfamily Caesalpinioϊdae, the family Fabaceae, part of the order Fabales (Rosales), class Magnoliopsida [20]. For many years, the genus *Ceratonia* L. was considered monotypic, including only the species *C. siliqua*, which is now found in all coastal regions of the Mediterranean [21]. In 1980, Hillcoat, Lewis, and Verdcourt described two subspecies: subsp. *oreothauma*, from Arabia (Oman), and subsp. *somalensis*, native to northern Somalia. *C. oreothauma* is morphologically very distinct from *C. siliqua*. Furthermore, *C. oreothauma* has slightly smaller pollen grains compared to *C. siliqua*, and they exhibit a tricolporate structure rather than a tetracolporate one [3]. Given that tetracolporate pollen grains represent a more evolved form than tricolporate grains, it has been suggested that *C. oreothauma* could be the wild ancestor of the cultivated *C. siliqua* [22].

## 4. Morphological Description of *C. siliqua* L.

### 4.1. Tree

The carob tree is a drought-resistant, perennial tree that produces for an extensive period (100 to 150 years) [17]. It showcases a very spreading and rounded crown with a thick, heavily cracked, and twisted trunk, similar to an olive tree. This is because the carob tree also grows slowly and has a long lifespan, potentially up to 500 years. The base of the trunk can reach 2 to 3 m in circumference. This woody species has smooth, gray bark when young that turns brown and rough as it matures. Its wood is yellowish white in its early stages, becoming veined pink then dark red and hard with age. This tree develops a pivoting root system that can reach a depth of 18 m [23,24]. Its leaves are evergreen, quite large (10 to 20 cm long) and paripinnate, composed of 4 to 8 leaflets, rarely more. These leaflets are oval, whole, leathery, dark green and smooth on top, and paler underneath [25,26]. Carob trees shed their second-year leaves in early summer, renewing leaves in the spring. The deep, robust taproot penetrates deeper soil layers, absorbing the nutrients and water the plant needs. Figure 1 shows a photo of the tree of *C. siliqua*.

The tree undergoes a cyclic vegetative growth with two flushes typically occurring in spring and autumn, the latter of which is usually weaker or sometimes absent altogether [27,28]. Vegetative growth decreases at lower temperatures (≤10 °C), but the intensity and onset of shoot growth in spring also depend on the bearing status of a tree or branch [29]. In fruit-bearing trees (known as “on” status), vegetative growth is delayed and diminished. In contrast, in non-fruit-bearing trees (or “off” status), the flushes are linked with higher levels of shoot growth. Single leaves are typically shed after 2 years in summer (July) and partially renewed in spring [30].

*C. siliqua* L. is dioecious, with a hermaphroditic morphology. The flowers are initially bisexual, but later removal of the organs renders them functionally unisexual. Dioeciousness is rare in Fabaceae. The flowers are small (6–12 mm long), with many green and reddish flowers, which multiply five-fold in late autumn on short peduncles with sepals but no corolla. The inflorescence usually comprises 15–40 individual flowers in short lateral racemes, mainly blooming on branches and stems older than 2 years (cauliflower and branch flowers). Flower characteristics can vary greatly depending on the cultivar [29]. The number and density of flowers in female inflorescences are lower than those in male and hermaphrodite inflorescences [31,32]. The flower consists of a pistil (6–8.5 mm) and a rudimentary stamen enveloped by five hairy sepals, with a curved ovary comprising two carpels (5–7 mm long) and numerous ovules. The stigma is made up of two lobes. The male flower consists of a nectary disc with five delicate filamentous stamens surrounded by hairy sepals. Hermaphroditic flowers include a pistil and five stamens [19].

### 4.2. Fruit and Seed

The fruit manifests as a non-dehiscent pod, slender, compressed, straight or curved, and thickened at the suture. It is 10–30 cm long, 1.5–3.5 cm wide, with a blunt or subacute apex, about 1 cm thick. The pods turn a brown color upon ripening, featuring a wrinkled and leathery surface. The pulp consists of a hard outer layer (the pericarp) and a softer inner region (the mesocarp). The seeds are arranged laterally in the pods, separated by the mesocarp. There are several seeds (4–15 seeds), typically uniform, smooth, shiny, brown, and oval-oblong (8–10 cm long, 7–8 mm wide, and 3–5 mm thick), difficult to inspect. In general, the seeds of the grafted varieties constitute 8 to 12% of the pod weight, whereas those of the ungrafted wild varieties represent 16 to 20% of the pod weight [19].

## 5. Origin and Geographical Distribution

Most researchers consider the carob tree to be native to the eastern Mediterranean basin [33,34,35], although some believe it originated elsewhere. Schweinfurth (1894) suggests that the carob tree is native to southern Arabia (Yemen) [2] while Zohary (1973) considers it to be part of the flora of Indo Malaysia, along with *Olea*, *Laurus*, *Myrtus*, and other plants [36]. Additionally, *C. oreothauma*, the only known species related to the carob tree, is native to southeast Arabia, specifically Oman, and the Horn of Africa, particularly northern Somalia [3].

In terms of climate, the original centers of the Caesalpinoideae subfamily were initially hot and humid, but after the Cretaceous period, the land was largely drained and uplifted, resulting in a cool, dry, and even desert climate. Other caesalpinioid legumes are primarily tropical and subtropical [37]. Hence, the carob tree seems to have evolved in a climate different from the Mediterranean [38]. The carob tree grew wild in Turkey, Cyprus, Syria, Lebanon, Palestine, southern Jordan, Egypt, Arabia, Tunisia, and Libya before reaching the western Mediterranean. It was disseminated by the Greeks in Greece and Italy and by the Arabs along the northern coast of Africa and the south and east of Spain. Since then, it has spread to the south of Portugal and south-east of France [3]. The Spaniards later introduced it to South and North America and Australia. Currently, the carob tree is also found in the Philippines, Iran, South Africa, and India [39]. In Morocco, the carob tree is both spontaneous and cultivated at the thermo-Mediterranean level. It has a strong presence in the meso-Mediterranean region with a semi-arid to sub-humid bioclimate, absolute minima above 3 °C, and altitudes from 0 to 500 m. Rarely does it reach 900 m or even 1600 m [40]. It grows in association with the olive tree, lentisk, cedar, or argan tree, in the plains and the middle mountains of the Rif, the Middle Atlas, the High Atlas, and the Anti-Atlas. The main spontaneous carob population is found in the regions of Tafechna and Aît Ishaq (province of Khenifra), between 600 and 1000 m above sea level, along with other forest species, sheltered from the winds and cold [41]. It plays a significant socioeconomic role, generating a total revenue that would be at least 7.94% of the total value of crop production for farmers in the north of the country [42]. Due to its adaptability to soil and climate stress, it could contribute to the development of marginal areas with an arid or semi-arid climate, rugged terrain, poor soil, and where specific erosion is high, and the regression of the vegetation cover limits any development of these zones [43].

## 6. Ecology

The carob tree belongs to the “Mediterranean coastal maquis” ecosystem and typically grows on limestone soils. It is a thermophilic, xerophilic, and heliophilous species. It thrives in temperate and subtropical regions and can tolerate hot and humid coastal areas. However, it is quite sensitive to winter cold, and hence, it is primarily found near coasts with altitudes below 500 m. In some regions, its cultivation expands to higher elevations, given suitable exposure [44].

The carob tree is a species that exhibits strong resistance to drought [11]. Studies confirm that the carob tree behaves like a true drought-resistant species, adapting morphologically and physiologically to water scarcity [44]. Owing to its ability to adapt to soil and climate stress, the carob tree could contribute to the development of disadvantaged areas [42], currently referred to as “grey areas”. Additionally, it is characterized by its high tolerance to salinity, able to tolerate up to 2 g/L of NaCl, making it more resilient than the date palm, pistachio, and olive trees, among others [44]. A study by Correia et al., 2010 [45] revealed that the carob tree can tolerate and maintain most of its physiological processes at a NaCl concentration of 2.32 g/L. This suggests that the carob tree can play a crucial role in saline areas. The carob tree, along with *Pistacia lentiscus* L. and *Olea europaea* L. var. *sylvestris*, constitutes one of the most characteristic associations of the lower zone of Mediterranean vegetation, and is thus considered a climax community (Oleo-Ceratonion) [2].

## 7. Production Areas

As mentioned earlier, carob production has a long tradition, mainly for animal feed, in most countries bordering the Mediterranean Sea. It is believed that the Greeks and Arabs were responsible for spreading carob throughout the Mediterranean basin from the Near East. More recently, carob has been introduced into warm, semi-arid zones of Australia, California, Arizona, Chile, Mexico, South Africa, and others. However, commercial carob production remains concentrated in the Mediterranean region. The total cultivated area of carob in the world is estimated at 87,485 ha, of which 74,174 ha (84.81%) are distributed between Spain, Morocco, Italy, and Portugal. Pod and seed production vary across different countries due to differences in seed yields between cultivars and wild-type varieties [2].

According to the Food and Agriculture Organization Corporate Statistical Database (FAOSTAT), the annual global production of carob was estimated at 51,906.58 tons in 2021, derived from 14,722 hectares primarily located in Europe (59.5%) and Africa (21.6%). The largest production is recorded in Portugal. Specifically, Portugal, Italy, Morocco, Greece, Turkey, and Cyprus are the biggest carob producers, having produced on average 39,935.93; 30,744.23; 22,436.42; 14,696.18; 14,337; and 6689.21 tons, respectively, during the period from 1994 to 2021 [46]. Conversely, Moroccan production has increased over the past 30 years, estimated at 26,000 tons in 2006. The most productive regions include Fez, Marrakech, Agadir, Essaouira, Taza, El Hoceima, Khenifra, and Beni Mellal [11]. In Morocco, carob tree cultivation relies primarily on rainfall. Production is estimated at 16,000 tons in pods (4800 tons in seeds), having consistently increased over the past fifteen years. Density per hectare ranges from 5 to 25 trees. The carob tree is a highly adaptable, heliophilous, heat-tolerant species, very resistant to drought (with 200 mm of rain) but not to cold. It tolerates poor, sandy, heavy loamy, rocky, and calcareous soils, with a pH range from 6.2 to 8.6, but it does not fare well in acidic and humid soils [41]. Table 1 below shows the carob production in three Maghreb countries in 2021.

## 8. Applications of Carob

### 8.1. Traditional Uses

Humans have been using carob as a food source and for medicinal purposes since antiquity, due to its edible fruits, commonly referred to as pods or merely carob. Currently, the primary use is the extraction of gums from the seeds. The use of carob dates back to the ancient Egyptians, who fed livestock with carob pods and reputedly used the gum as an adhesive in mummy binding. The Arabs used the carob seed as a unit of weight, referring to it as “qirat” or “karat”. The standard weight of the carob seed became the unit of weight for gold and precious stones [4]. The fruit of the carob tree consists of a pulp enveloping regular seeds. Indeed, the sweet pulp of carob has long been used as livestock feed alongside other foods, such as barley flour. It seems that carob flour is particularly suited to piglet feed [25].

The carob tree, *C. siliqua* L., is renowned for its medicinal properties. Indeed, carob pods have been reported to exert anti-inflammatory, antimicrobial, antidiarrheal, antioxidant, antiulcer, anti-constipation, and glucose absorption-inhibiting activities in the gastrointestinal tract [4]. Traditionally, carob fruits have also been used as an antitussive and against warts [47,48]. *C. siliqua* L. has been used to treat several diseases. Among its traditional uses in integrative medicine, we find that in Palestine, carob pods and seeds are used to treat high blood pressure [7]. An infusion of carob leaves is used as an emetic in cases of acute intoxication [49]. In Tunisia, the carob tree is utilized for treating gastrointestinal disorders, and carob juice is also used to address diarrhea [50]. In Morocco, carob leaves and seeds are employed in the treatment of diabetes [51,52]. Carob leaves have traditionally been used in Turkey to remedy diarrhea [44]. Carob wood was also traditionally used to make slow-burning charcoal [2] (Table 2).

### 8.2. Industrial Applications

The carob tree (*C. siliqua* L.) holds potentially significant importance for the food industry due to its chemical constituents, flavoring properties, and nutrition benefits. It is also exploited in the fields of cosmetology, chemistry, and medicine. Thriving in the semi-arid growing conditions of the Mediterranean region, the carob tree has an annual worldwide production of over 315,000 tons of carob products [60]. The carob bean consists of 90% pulp and 10% seeds, the latter being obtained after breaking the carob pods. Characterized by a brown color, significant hardness, a length of about 10 mm, and a weight of about 0.2 g per seed, the carob seeds are often left over as a by-product or food waste after the manufacturing of primary carob products such as flour, powder, and syrup [71].

The carob tree emits a strong characteristic aroma, which persists even after processing. This unique property may be attributed to the presence of acids, esters, and aldehydes/ketones emitted from carob fruit and powder, which are biogenic volatile organic compounds contributing to plant growth, breeding, and defense [72]. Vladimir et al., 2018 [73] studied the economic impact of incorporating plant ingredients into the innovative production of flour-based functional foods. The production of a lecithin bun with 4% carob resulted in an increased content of proteins, fat, and fibers and a decreased carbohydrate content, with a higher production efficiency than that of lecithin buns without carob. The price of the carob bun is only 6% higher than the one without carob addition. This analysis recognizes carob bean as an excellent raw material to produce gluten-free bread and flours enriched with vitamins, minerals, and proteins. Currently, carob pods are commonly used in cakes, cookies, drinks, and various snacks in Egypt. Jams and liquors are made from carob in Turkey, Malta, Portugal, Spain, and Sicily. In Libya, a syrup named “rub” is extracted from carob and used to make “asida”, a traditional dessert [4].

The sweet pulp of the carob has long been used as livestock feed, often combined with other foods such as barley flour [41]. The flour, obtained by drying, roasting, and grinding the pods after stripping them of their seeds, is used primarily in the food industry [74], in the preparation of sweet juices, chocolates, biscuits, and as a cocoa substitute [39]. Locust bean gum, extracted from the albumen of *C. siliqua* seeds due to its richness in galactomannans (units of β-D-mannose and α-D-galactose) from the endosperm, constitutes one-third of the total weight of the seed [75]. This gum is used in the agro-food industry as a thickener, known under the standardized code E410, and is widely applied in frozen dairy products, commercial sauces, dressings, dips, and confectionery. It is also used in cosmetics to form emulsions and shaving foams, pharmaceuticals (mainly against diarrhea), and in dietetic food preparations to reduce food intake in the treatment of obesity. Locust bean gum also has applications in printing, photography, plastics, ink, shoe polish, as a substitute for pectin, gelatin, as a food stabilizer, for bacterial growth, and other textile applications. In the field of chemistry, it is applied as glue, a polishing product, a match, and against pesticides. It also has other industrial uses as a flotation product, a moisture absorbent, a water scavenger for explosives, and in surface treatment [11,25,75]. Carob powder from the pods is a natural sweetener that tastes and looks like chocolate, which is why it is often used as a cocoa substitute. The advantage of using carob is that, unlike chocolate, it does not contain stimulants because it is devoid of caffeine and theobromine [76].

Currently, carob is being investigated as a source of new natural antioxidants, specifically those found in the seed coat and the pulp of the fruit. This antioxidant activity is attributed to the presence of phenolic compounds and fibers [77]. Carob flour (ground pods), obtained by drying, roasting, and grinding the pods after stripping them of their seeds [78], is used for sugar extraction, ethanol fermentation, and citric acid production. Likewise, carob wood is highly valued in cabinetmaking and for making charcoal, while the bark and roots are used in tanning [79].

## 9. Phytochemistry of *C. siliqua* L.

Phytochemical compounds in carob vary greatly, influenced by factors including environment, maturation stage, and tree parts [80].

In a study conducted by Fadel et al., 2011 [81], the phenolic compounds present in the aqueous acetone extracts of carob pulps and seeds were analyzed. These carob samples were harvested from two areas, Izouika and Reggada, in southwest Morocco. The High-Performance Liquid Chromatography (HPLC) analysis of the extracts showed their richness in phenolic compounds. In both areas, the pulp extracts were richer in phenolic compounds than the seeds, both qualitatively and quantitatively. The phenolic profile of the pulp was dominated by coumaric acid (20.52% in Izouika vs. 17.05% in Reggada) and gallic acid (17.8% in Izouika vs. 12.57% in Reggada). In seed extracts, coumaric acid and gallic acid were also the predominant phenolic acids, with coumaric acid representing 8.07% in Izouika and 8.18% in Reggada, while gallic acid represented 5.01% in Izouika and 3.95% in Reggada. Syringic acid, 4-hydroxybenzoic acid, and gentisic acid are all benzoic acids present in carob [82].

HPLC methods have been used to determine polyphenols in carob pods, revealing the presence of condensed tannins (proanthocyanidins), composed of flavan-3-ol groups and their galloyl esters, gallic acid, catechin, epicatechingallate (ECG), epigallocatechingallate (EGCG), and quercetin glycosides [83,84]. The presence of hydrolysable tannins (gallotanins and ellagitannins) has also been detected in carob pods [85]. Carob fiber was found to contain a rich variety of phenolic compounds, with a total of 24 polyphenol compounds identified, yielding 3.94 g/kg (dry weight). The profile was dominated by gallic acid in various forms: free gallic acid (42% of polyphenols by weight), gallotannins (29%), and methyl gallate (1%). Simple phenols, mainly cinnamic acid, comprised about 2% of the total, and major flavonoids were identified as the glycosides myricetin and quercetin-3-*O*-α-L-rhamnoside [80]. In another study, HPLC analysis showed the main compounds in mature carob pods to be pyrogallol (48.02 ± 3.55%), catechin (19.10 ± 2.11%), and tannic acid (9.01 ± 1.40%) [9]. In immature carob pods, pyrogallol (26.45 ± 3.03%), catechin (16.52 ± 2.34%), gallic acid (15.12 ± 2.31%), chlorogenic acid (15.01 ± 1.72%), and epicatechin (12.26 ± 1.04%) were detected [10]. The HPLC technique also identified several phenolic compounds in leaves such as kaempferol (77 ± 2.43%), tannic acid (13 ± 0.45%), catechin hydrate (4.30 ± 0.34%), and polydatin (0.85 ± 0.22%) [86].

Phytochemical screening of the crude ethyl acetate and methanolic extracts of three types of carob tree barks (spontaneous male, spontaneous female, and grafted female) indicated the presence of flavonoids and tannins. Alkaloids and saponins were not detected. The total phenolic contents from the ethyl acetate and methanolic extracts of the three varieties of *C. siliqua* L. barks varied from 0.46 to 0.76 (g/L gallic acid equivalents). In this study, the methanolic extract had a higher phenolic content than the ethyl acetate extract. These phenolic compounds and other reported bioactive compounds are generally more soluble in polar solvents [87].

Carobs are particularly rich in flavonols such as quercetin, myricetin, kaempferol, and their glycosidic derivatives. Quercetin and myricetin rhamnosides are usually the most abundant flavonoids in carob. The presence of flavones (apigenin, luteolin, and chrysoidium), flavanones (naringenin), or isoflavones (genistein) is low [80,83,88]. According to a semi-quantitative ultra-performance liquid chromatography (UPLC) analysis carried out on the leaves of *C. siliqua* from the southern region of Morocco (Tafraoute), it was found that the major compounds in the aqueous extract of *C. siliqua* were luteolin-7-glucoside followed by epicatechin, apigenin-7-glucoside, quercetin-3-*O*-glucoside, caffeic acid, gallic acid, and chlorogenic acid. This indicates that *C. siliqua* leaves represent a good source of natural bioactive compounds. No aglycons were identified in the sample [89] (Figure 2).

Indeed, carob pulp is a substantial source of dietary fiber and sugars, along with a variety of bioactive compounds including polyphenols, as previously mentioned. Dietary fiber is composed of various substances and is typically categorized as soluble or insoluble. Both types of fibers have important health benefits. In the context of carob:-Total dietary fiber in carob pulp ranges from 30 to 40% [90].-This fiber is primarily made up of lignin (50–65%), cellulose (15–25%), and hemicellulose (15–25%), with smaller amounts of pectin (0.5–2%), tannins (3–7%), and moisture (4–8%) [91].-Carob fiber is primarily insoluble and is not easily fermented by gut bacteria [92].-The soluble fiber content, which can be fermented in the colon, is significantly lower in carob and this portion contains simple carbohydrates [82].

The total sugar content in carob pulp typically ranges from 30 to 60%. Sucrose forms the largest portion of these sugars, contributing 65–75% of the total sugar content. The remainder of the sugar content is mostly made up of the monosaccharides fructose and glucose, contributing 15% and 25% of the total sugars, respectively [93]. Carob pods are especially notable for their high sugar content, which is even higher than that of beets or sugar cane (about 200 g/kg) [94]. Carob seeds, on the other hand, primarily contain the sugars sucrose (8.1 ± 0.04%) and glucose (2.2 ± 0.01%). They also contain various monosaccharides following polysaccharide hydrolysis, namely mannose (54.0 ± 0.50%), galactose (15.5 ± 0.14%), glucose (2.2 ± 0.01%), arabinose (1.0 ± 0.00%), and xylose (0.4 ± 0.00%). Fructose was not detected [95].

Carob is a good source of protein, and its protein content is particularly rich in both essential and non-essential amino acids. The protein fraction in carob seeds, for instance, has a substantial amount of the non-essential amino acids like arginine (27.8 ± 0.25 g/100 g) and alanine (17.0 ± 0.16 g/100 g) and the essential amino acid lysine (15.0 ± 0.14 g/100 g). Moderate amounts of essential amino acids like isoleucine (8.6 ± 0.08 g/100 g) and valine (7.3 ± 0.06 g/100 g) are also present in carob seeds [95]. In terms of the overall amino acid content in carobs, it is composed of a mixture of 17 residues including aspartic acid, glutamic acid, serine, glycine, histidine, arginine, threonine, alanine, tyrosine, valine, proline, methionine, isoleucine, leucine, cysteine, phenylalanine, and lysine [96,97]. In particular, aspartic acid, asparagine, alanine, glutamic acid, leucine, and valine make up approximately 57% of the total amino acid content of the pods [98]. Ground carob flour contains about 4.45% protein, with carob germ flour proteins including albumin and globulin (32%) and glutelin (68%) [96]. Importantly, no prolamins are detected in carob germ flour proteins [99], which contributes to the gluten-free property of carob [100]. Carob seeds are characterized by high protein content (25.7 ± 0.18%). Even in naturally grown carob seeds, like those from the Tazmalt region in northern Algeria, a moderate protein content was reported (18.6 ± 0.3%) [101]. Carob leaves also have a balanced nutrient composition. A proximate analysis based on dry weight showed the presence of carbohydrates (16.62%), fat (4.60%), protein (22.25%), ash (4.255%), and dietary fibers (11.77%) [102].

According to research by Özcan et al., 2007 [103], the protein, oil, crude fiber, ash, and energy values of carob fruit and carob flour were found to be not statistically different. However, carob syrup showed lower values for protein, crude fiber, ash, and energy compared to both carob fruit and carob flour. Interestingly, carob syrup had the highest total sugar content, with 48.3% in the fruit and 41% in the flour. Carob pods from the eastern parts of Sicily in Italy, as reported by Avallone et al., 2002 [85], had the following composition (dry weight): moisture (6–10%), ash (1–6%), protein (1–5%), fat (0.4–0.8%), sucrose (27–40%), D-glucose (3–5%), D-fructose (3–8%), and starch (0.1–1.3%). The carob germ which consists of fine fragments of hull and endosperm, and could be obtained industrially, has the following composition: 8.3% moisture, 6.5% ash, 6.6% lipids (neutral and polar) which contain approximately 21% polar lipids, 54.7% crude proteins, and an energy value of 17.5 kJ/g [104]. Moreover, the average proximate composition of raw carob pods is as follows: 8.17–9.56% moisture, 89.57–91.12% carbohydrates, 40.69–54.74% total sugars (33.70–45.09% sucrose, 1.79–4.95% glucose, and 1.80–5.19% fructose), 29.88–36.07% dietary fiber, 3.07–4.42% protein, 2.58–3.08% polyphenols, 0.45–0.86% fat, and 2.13–2.69% ash [17].

The mineral content of carob varies depending on the part of the tree analyzed (leaves, bark, integument, pulp) and the specific variety of the carob tree (*C. siliqua* L.), such as spontaneous female, spontaneous male, and grafted female. In a study conducted in Chefchaouen, northern Morocco, it was found that calcium (Ca) generally appears to be the most prevalent mineral in the leaves and bark of the carob tree, regardless of the category studied. Other detected elements in significant amounts include potassium (K), magnesium (Mg), sodium (Na), phosphorus (P), chlorine (Cl), copper (Cu), iron (Fe), zinc (Zn), and selenium (Se). It was also demonstrated that while potassium (K) was the main macronutrient in the bark, leaves had higher concentrations of other minerals such as calcium (Ca), magnesium (Mg), phosphorus (P), and zinc (Zn). Conversely, for copper (Cu), iron (Fe), and selenium (Se), the bark had higher levels [105]. A separate study of Anatolian carob pods by Ayaz et al. (2009) discovered the following mineral concentrations: 970 mg/100 g K, 71 mg/100 g P, 300 mg/100 g Ca, 60 mg/100 g Mg, 1.88 mg/100 g Fe, 1.29 mg/100 g manganese (Mn), and 0.85 mg/100 g Cu [98]. Furthermore, research on both wild and grafted carob trees in the province of Antalya in Turkey showed that the pods and seeds of the grafted carob trees generally had higher mineral concentrations than their wild counterparts. In all the samples, potassium was the most prevalent mineral, with the highest concentration found in the pods of the wild carob fruit. Of the micro-minerals, iron was the most abundant in the seeds of grafted carob fruits [106].

The study by El Bouzdoudi et al., 2017 [107] on carobs grown in Morocco identified five macroelements (potassium, calcium, chlorine, magnesium, and sodium) and thirty microelements (iron, aluminium, strontium, rubidium, copper, zinc, manganese, zirconium, barium, thorium, bromine, thulium, chromium, selenium, cerium, lanthanum, caesium, antimony, neodymium, mercury, cobalt, arsenic, scandium, molybdenum, tantalum, gold, hafnium, samarium, silver, and europium) in the whole pod, pulp, and seed and other constituents. The detection of these mineral elements in such a wide variety underscores the role of carob as a valuable source of essential nutrients, contributing to its longstanding use in human and animal diets. Moreover, the study by Khlifa et al., 2013 [93] examining carob pods from the Chefchaouen region in northern Morocco indicated a low-fat content in the carob pod. Another study on carob seeds from Malaga, Spain, which involved treating the seeds with acid or hot water to isolate the carob germ meal, revealed the presence of polar lipids accounting for 13–24% of the total lipid content (1.1–1.7% of carob germ meal). This lipid fraction primarily consisted of oleic acid (34.4%) and linoleic acid (44.5%) as the major fatty acids, with palmitic acid (16.2%) and stearic acid (3.4%) being the main saturated fatty acids [104]. According to Fidan et al., 2020 [95], the lipid fraction content of carob seeds was determined to be around 2.1%, consisting of non-saponifiable lipids (17.2 ± 0.13%), sterols (4.0 ± 0.04%), phospholipids (7.2 ± 0.07%), and tocopherols (2801.0 ± 40.16 mg/kg). They identified 11 fatty acids that constituted the total oil content, with oleic (45.0 ± 0.42%) and linoleic (32.4 ± 0.30%) acids being the most abundant. Palmitic (16.6 ± 0.15%) and stearic (4.7 ± 0.15%) acids were also present in moderate amounts. Research by Dallali et al., 2018 [108] on Tunisian carob tree leaves found a fatty acid profile dominated by unsaturated fatty acids. Among the 12 common fatty acids identified, linolenic and linoleic acids were the most prevalent. Table 3 groups the chemical compounds present in the different parts of the carob tree including the leaves, pods, pulp, and seeds.

## 10. Pharmacology of *C. siliqua* L.

### 10.1. Antioxidant Activity

Numerous studies point towards the considerable antioxidant activities of various extracts from C. siliqua, including leaves, seeds, and kibble.

The study by Ouahioune et al., 2022 [136] focused on carob leaves (CL), seeds (CS), and kibble (CK), showing that all three have considerable antioxidant activities by the DPPH (2,2-diphenyl-1-picrylhydrazyl) method. However, there was a clear order of potency; it decreased in the following order: CSE > CSA > CLE > CLA > CKE > CKA. Their respective total antioxidant capacities were 72.81 ± 0.09; 67.22 ± 3.91; 53.52 ± 0.09; 51.76 ± 0.20; 26.96 ± 2.14; and 12.35 ± 0.46 μg/mL (expressed as ascorbic acid equivalents). The following abbreviations of macerates were applied: CLE, CSE, and CKE, for leaves, seeds, and kibbles, respectively, macerated in 80% ethanol, and CLA, CSA, and CKA for leaves, seeds, and kibbles macerated in 80% acetone. In another study that focused on carob leaves from various types of trees, including spontaneous male, spontaneous female, and grafted female, located in the Province of Chafchaouen (NW of Morocco), it was observed that ethyl acetate extracts exhibited the most potent scavenging effect on the DPPH radical. This was in comparison to the diethyl ether (Et_2_O), dichloromethane (CH_2_Cl_2_), and ethyl acetate (EtOAc) extracts. Thus, the lowest activity was found in the CH_2_Cl_2_ extract. The DPPH radical scavenging activities of the plant extracts ranged from 1.17 to 61.17%. The results of the analyses of the EtOAc extract demonstrated that the most active radical scavengers were found in the grafted female category (IC_50_ = 0.41 g/L) followed by the spontaneous female category (IC_50_ = 0.45 g/L) and the spontaneous male category (IC_50_ = 1.50 g/L) [137]. Carob contains large amounts of polyphenols with strong antioxidant and antiradical power as well as reducing properties that can be an important source of natural antioxidants. A study by Saci et al., 2020 [138] demonstrated that carob pulp extracts exhibit significant free radical scavenging activity, evaluated using the ABTS assay. The highest activity was observed from the wild variety in the unripe stage (IC_50_ = 3.39 ± 0.25 μg/mL). Another study by Rtibi et al., 2017 [139] revealed that the radical scavenging activity of extracts against ABTS in immature carob pods increased significantly in a dose-dependent manner, with an EC_50_ of 175.41 ± 3.04 μg/mL. The antioxidant activity of this tree is attributed to its potent scavenging effect on reactive oxygen and free radicals as well as its ability to inhibit myeloperoxidase (MPO) activity in a concentration-dependent manner [9,86]. An investigation by Lakkab et al., 2019 [135] that focused on determining the reducing power (FRAP) found that the MeOH and EtOAc fractions of the carob seed peel extracts (CSP) exhibited the most potent reducing effects. They attributed the significant reducing power to the polyphenol and flavonoid content of the extracts and identified that most of these compounds were found in the acetone extract. The total phenol content and antioxidant activity of carob flour extracts, evaluated using the FRAP assay, revealed that the total phenol content for carob flour is 17.7 mg/g and the antioxidant activity obtained through this method was 249.2 ± 28.8 µmol Fe^2+^/g [140].

Scavenging free radicals is among the primary antioxidant mechanisms that inhibit lipid peroxidation chain reactions and reduce the harmful effect of cytotoxic products. Thus, carob extract has been shown to influence SOD and catalase activities, as reported in a rat model, which could enhance their performance and ability to eliminate ROS [50]. In another study, the antioxidant activity of the crude polyphenolic fraction of carob pods (CPP) was evaluated using the β-carotene bleaching test. The concentration of 10 μg/mL of CPP exhibited antioxidant capacity equivalent to (−)-epicatechin gallate, (−)-epigallocatechin gallate, and quercetin [141].

These findings contribute to the understanding of the antioxidative properties of carob extracts, highlighting their potential in preventing or mitigating oxidative stress-related disorders. This potent antioxidant activity is largely attributed to the high content of polyphenolic compounds present in carob extracts. These results reinforce the potential of carob as a valuable source of natural antioxidants, which can be important in the diet for maintaining health and preventing disease.

### 10.2. Antibacterial and Antidiarrheal Activity

The antibacterial activity of a methanol extract of *C. siliqua* fruits was tested and compared to a methanolic extract of *Plantago major*. The carob fruit extract showed higher activity against most bacteria, including *Enterococcus* spp. Both aqueous and methanolic extracts were evaluated for their antibacterial activity individually and in combination with other antibacterial agents, such as ampicillin, gentamicin, amikacin, and clindamycin. The combination of the extract and antibacterial agent proved to be more effective than each one used separately. The extracts were analyzed, and several compounds demonstrating antibacterial activity were identified and characterized. The ethanol and acetone extracts were tested for antibacterial activity against *Pectobacterium atrosepticum* in potato soft rot, and the acetone extract exhibited greater activity than the methanol extract. A methanolic leaf extract was found to be effective against *Listeria monocytogenes*. HPLC analysis of the extract revealed seven compounds with antibacterial activity, especially epigallocatechin-3-gallate [122].

The results of in vivo studies indicated that *C. siliqua* leaf extract had significant antidiarrheal effects due to its inhibitory effect on gastrointestinal secretion and electrolytes. This suggests that the aqueous extract of *C. siliqua* leaves may hold potential for the development of antidiarrheal drugs [142]. Recent years have seen studies into various multifactorial gastrointestinal problems, including constipation and diarrhea. Research has been conducted into the potential effects of the aqueous extract of carob pods on gastrointestinal transit (GIT), diarrhea, and intestinal epithelial permeability in healthy rats and mice. It was observed that the aqueous extracts of ripe carobs significantly increased GIT in a dose-dependent manner. Conversely, the aqueous extract of immature carobs significantly reduced GIT. The differing effects were attributed to variances in the chemical composition of the two extracts. The extract from ripe carob pods is high in fiber and sugar, compounds that are either absent or present in small amounts in immature carob pods. It has been suggested that high concentrations of fiber and sugar can accelerate the GIT process. The high content of total tannins in immature carob pods may account for the observed GIT inhibition and reduction in diarrhea [4]. Several clinical studies have highlighted the effectiveness of carob powder in treating acute childhood diarrhea [143]. Carob fruits are rich in pectin and tannin, and the seeds are high in mannans and galactans. These compounds have potent antidiarrheal properties and are commonly used in the treatment of infant diarrhea. Carob bean gum, a food thickener, may also be useful in the treatment of infantile gastroesophageal reflux [39].

Hexane, ethyl acetate, chloroform, and methanol extracts of dried carob leaves were prepared and tested for antifungal activity against *Geotrichum candidum*, the agent responsible for citrus sour rot. Hexane and chloroform extracts were less active, while the methanol extract exhibited more activity than the ethyl acetate extract [144]. Carob essential oil possesses antimicrobial, cytotoxic, and other biological properties, making it of potential interest to the medical and pharmaceutical fields [145].

These studies suggest that the various extracts of *C. siliqua*, including methanolic, aqueous, ethyl acetate, and others, have displayed substantial antibacterial and antifungal properties. Moreover, they also demonstrated remarkable efficacy against gastrointestinal disorders, notably in modulating gastrointestinal transit and reducing diarrhea.

### 10.3. Anti-Inflammatory and Antiulcer Activity

A study aimed to determine the anti-inflammatory potential of methanolic bark extracts of *C. siliqua* in rodents (female Swiss mice and male Wistar rats). The methanolic bark extract of carob was studied for its anti-inflammatory properties using carrageenan and experimental trauma-induced hind paw edema in rodents. The results obtained showed that the methanolic bark extract of *C. siliqua* significantly reduced and inhibited edema, with effects comparable to those of the control and reference drugs used in both models [49].

The antiulcerogenic effect of methanolic crude extract of *C. siliqua* was examined using three ulcerogenic models: HCl/ethanol, pyloric ligation, and aspirin. The results showed that the crude extract of *C. siliqua* has significant gastroprotective and antisecretory properties that prevent gastric ulceration induced by various necrotizing agents. In the HCl/ethanol-induced ulcer model, oral administration of the HCl/ethanol solution to the control group produced characteristic necrotizing mucosal lesions confined to the glandular portion with an ulcer index of 2.26 ± 0.44. Ulcer formation was significantly reduced in rats pretreated by *C. siliqua* with a percentage inhibition of ulcer formation of 50.7%. In the pyloric ligation model, ulcers develop due to the accumulation of gastric acid and pepsin, which leads to autodigestion of the gastric mucosa. The ulcer index was significantly reduced in animals pretreated by the plant. The gastroprotective effect of the plants was substantial, with up to 79.8% protection against ulcers. The volume of gastric secretions in pretreated rats was significantly reduced by *C. siliqua* by 46.4%. The plant also significantly reduced the volume of gastric juice, similar to the effect of omeprazole, a proton pump inhibitor. The antiulcer activity of the studied plant was also evaluated using aspirin as an ulcer-inducing model. The results showed substantial protection against ulcers in tests with positive controls, cimetidine, and omeprazole. The methanolic extract of *C. siliqua* demonstrated an ulcer-inhibiting effect of 51.9% in this model [146]. Rtibi et al., 2016 [10] showed that all parameters indicating the presence of inflammation induced by sodium dextran sulfate DSS (5%), such as the primary indicator myeloperoxidase (MPO) and cytokines (TNF-α and IL-1β), were inhibited by the aqueous extract of carob pods. This suggests that carob may have a potent anti-inflammatory effect on inflamed colonic tissue and plasma in rats.

Thus, *C. siliqua* extracts hold considerable potential as natural anti-inflammatory and gastroprotective agents, warranting further investigations.

### 10.4. Antihyperglycemic Activity

Minimization of glucose uptake could potentially aid in controlling hyperglycemia and could represent a novel mechanism for an antidiabetic agent in diabetic patients [147]. The hypoglycemic effect of the aqueous extract of carob leaves was recently studied by Rtibi et al., 2018 [148], who showed that the aqueous extract reduced glucose absorption in vivo and in vitro in a dose-dependent manner. Qasem et al., 2018 [149] evaluated the in vitro inhibitory effect of the methanolic extract of carob pods against amylase and glucosidase and the in vivo glycemic effect of this extract in diabetic rats induced by streptozotocin and nicotinamide. The methanolic extract showed clear inhibitory effects against both amylase and glucosidase. Moreover, a high dose of carob exhibited antihyperglycemic activity in vivo. Another work reports the in vitro inhibitory activity of the water decoction of leaves, germ meal, pulp, locust bean gum, and locust bean rind on α-amylase and α-glucosidase, with the leaf and stem bark decoction strongly inhibiting all tested enzymes [150]. In addition, another study by Rtibi et al., 2017 [139] showed that immature carob could prevent intestinal glucose absorption through the inhibition of electrogenic sodium-dependent glucose transport in mice, using the Ussing chamber technique. More importantly, at different doses, immature carob demonstrated a significant reduction in blood sugar and biochemical profiles in diabetic rats. D-Pinitol is a naturally occurring inositol found in many plants. Carob contains the highest content of D-pinitol, which has a wide range of medicinal and other properties. One of these, and perhaps the most important, is insulin regulation, which has two main mechanisms: insulin-sensitizing and insulin-mimetic activities [151].

The numerous studies outlined here collectively indicate the promising antidiabetic potential of carob extracts. These findings include evidence of reduced glucose absorption, inhibition of enzymes integral to glucose metabolism, and the beneficial regulation of insulin activity.

### 10.5. Anti-Obesity Activity

According to a study by Lasa et al., 2017 [152], designed to assess the effects of *C. siliqua* by-products on triglyceride (TG) accumulation in mature 3T3-L1 adipocytes, methanolic extractions were carried out on germ, seed bark, and pods. Mature 3T3-L1 adipocytes were treated for 24 h with these extracts at doses of 0.5, 0.1, and 0.05 mg/mL, and the TG content was quantified. The findings showed that the seed bark and pod extracts facilitated a reduction in TG in adipocytes at the 0.1 mg/mL dose. Notably, no cytotoxic effects were observed at any extract dosage in the adipocytes.

The methanolic extract of carob pods caused a reduction in the triglyceride content of mature 3T3-L1 adipocytes at a carob dosage of 100 mg/mL (28% reduction). Similarly, combinations of 1/5 and 1/50 (wakame/carob) decreased triglyceride content by 27% and 36%, respectively. Combinations of 1/5 and 1/50 wakame and carob pods in a snack formulation displayed fat reduction properties both in vitro and in vivo [153]. Similarly, Rico et al., 2019 confirmed that fat accumulation in mature adipocytes was reduced by the methanolic extract of carob seed peels and pods [154]. Another study demonstrated the anti-obesity effects of carob leaf infusion. The aqueous extract of carob prevented a high-fat diet (HFD)-induced body weight gain by modulating serum lipid profiles (triglycerides, total cholesterol) [155].

From carob leaf infusion to methanolic extractions of carob pods, there is compelling evidence pointing towards their ability to decrease lipid accumulation in adipocytes, modulate serum lipid profiles, and counterweight gain associated with high-fat diets.

### 10.6. Antiproliferative Activity

Extracts from the plant *C. siliqua* L. have been extensively studied for their potential to prevent many diseases, primarily due to the presence of polyphenolic compounds. A recent study explored the anticancer properties of Cypriot carobs for the first time. They produced extracts from ripe and unripe carobs, pulp, and seeds using solvents of different polarities. They measured the ability of the extracts to inhibit proliferation and induce apoptosis in immortalized normal and cancerous breast cells, using the MTT assay, cell cycle analysis, and Western blotting. The results show that the antiproliferative potential of carob extracts varies with the stage of carob maturation and the extraction solvent. Diethyl ether and ethyl acetate extracts derived from the ripe whole fruit had a high myricetin content and exhibited specific activity against cancer cells. Their mechanism of action involves caspase-dependent and independent apoptosis. Based on these results, they suggested that Cypriot carob extracts may have potential applications in the development of nutritional supplements and pharmaceuticals [123].

In a study conducted by Custódio et al., 2011 [156], the goal was to evaluate the antiproliferative and apoptotic potential of methanolic extracts of carob leaves and pulp in human breast cancer. Cells were treated with methanolic extracts from leaves and pulp for 24, 48, and 72 h. The methanolic extract significantly inhibited cell proliferation in a dose-dependent manner, and the best incubation periods were 48 and 72 h. Leaf extracts showed a significantly higher ability to inhibit cell proliferation than pulp extracts. Treatment with leaf extracts resulted in an increase in cells at the sub-G1 stage (apoptotic cells). The results suggest that carob leaves and pulp could be a source of phenolic compounds with potential anticancer activity.

Carob pod and leaf extracts were tested for their ability to inhibit cell proliferation of the mouse hepatocellular carcinoma cell line (T1). Both extracts showed marked impairment of T1 cell proliferation in a dose-related manner, reaching maximum effect at 1 mg/mL. Furthermore, it was demonstrated that leaf and pod extracts were able to induce apoptosis in T1 cell lines after 24 h of treatment, inducing direct activation of the caspase 3 pathway. The finding that carob and carob leaf extracts contained antiproliferative agents may have practical importance in the development of functional foods and/or chemopreventive drugs [84]. Corsi et al., 2002 [9] evaluated the antiproliferative effect of carob pod and leaf extracts against the hepatocellular carcinoma cell line of mice (T1). Both extracts showed a remarkable deregulation of T1 cell proliferation in a dose-dependent manner and expressed the highest effect at the concentration of 1 mg/mL. Moreover, leaf and pod extracts induced apoptosis in T1 cell lines after 24 h of treatment, mediating a direct activation of the caspase 3 pathway. In another study, the pretreatment of experimental rats, by intraperitoneal injection for 8 days of the ethyl acetate extract of the leaves of *C. siliqua*, protected the rats against CCl_4_-induced hepatic and renal disorders. The biochemical parameters were consistent with the histological observations showing hepatoprotective and nephroprotective effects of *C. siliqua* [43].

These studies highlight the potential anticancer properties of extracts from various parts of the *C. siliqua* plant. Whether derived from the leaves, pulp, or pods, these extracts were found to inhibit cell proliferation and induce apoptosis in various cancer cell lines. They could potentially offer hepatoprotective and nephroprotective effects, further broadening their potential therapeutic applications. The key active compounds appear to be the polyphenolic compounds, specifically myricetin, which were found to trigger both caspase-dependent and independent apoptosis.

### 10.7. Antidepressant Activity

Acetone extract from fresh pods was prepared, analyzed for tannin content, and tested as an antidepressant with the tail suspension test and forced swimming test. It was active in both tests, and the proposed mechanism of action was through interaction with the adrenergic and dopaminergic systems [157]. A recent study was investigated on the effect of carob on mood disorders. The effect of carob seed peel (CSP) extracts on mood disorders and anxiety was accessed using the Y-maze, elevated plus maze, and forced swimming tests. The behavior test was primarily conducted using the ethyl acetate and acetone extracts of CSP. Ethyl acetate and acetone extracts of CSP were used to evaluate short-term memory and anxiolytic effect using Y-maze task. Both acetone and ethyl acetate extracts enhanced the spontaneous alternation in the Y-maze task and time spent with arms extended in the high plus maze task. However, the acetonic extract proved its efficacy in the forced swimming test [135]. Another study conducted by Avallone et al., 2002 [158] showed that *C. siliqua* pod and leaf extracts could be used to obtain anxiolytic and sedative effects.

These findings support the idea that extracts from carob pods and seed peels could have potential therapeutic benefits in the treatment of mood disorders such as anxiety and depression.

### 10.8. Antihypertensive Effect

A very recent study explored the pharmacological mechanisms involved in the bronchorelaxant and vasorelaxant activities of the pods of *C. siliqua* L. on isolated tracheal, aortic, and auricular fragments of rabbit. Additionally, a study on antihypertensive activity in normotensive rats was performed. The results demonstrated that the crude extract (Cr.Cf), the dichloromethane fraction (Dc.Cf), and the alkaloid fraction (Dc.Af) of *C. siliqua* relaxed the carbachol-induced contractions in the tracheal fragment. Cr.Cf completely relaxed the phenylephrine (PE)-stimulated contraction and partially relaxed the K^+^-mediated contraction in the rabbit aortic fragment. Dc.Cf followed the crude extract model, completely relaxing the PE-induced contraction and partially relaxing the K^+^-mediated contraction, while the Dc.Af fraction did not induce any relaxant effect on PE but showed partial relaxation of K^+^-mediated contraction. Cr.Cf induced a hypotensive response in terms of reduction of systolic blood pressure, diastolic blood pressure, as well as mean arterial pressure during its intravenous administration to normotensive anesthetized rats. The bronchodilator effect may be mediated by antagonism of muscarinic receptors, and they hypothesized that vasodilator and hypotensive effects are mediated by antagonism of α-adrenergic receptors and the involvement of calcium channel blockade as well as the agonist activities of cardio-selective muscarinic [159]. A study was carried out on the beneficial effects of a commercial carob fruit extract (CSAT+^®^) on cardiometabolic alterations associated with metabolic syndrome (MetS) in mice. Mice were fed for 26 weeks either with a standard diet or with a diet rich in fats and sugars, supplemented or not with 4.8% of CSAT+^®^. The results showed that CSAT+^®^ supplementation prevented MetS-induced hypertension and decreased the vascular response of aortic rings to angiotensin II. Additionally, treatment with CSAT+^®^ attenuated endothelial dysfunction [160]. A study focusing on the effect of white bean extract and *C. siliqua* extract in combination with a green tea extract in the treatment of excess weight and obesity was conducted. A test composition containing 200 mg of white kidney bean extract, 50 mg of locust bean gum extract, and 100 mg of green tea extract was administered twice a day, in capsule form half an hour before lunch and dinner. The effect of this composition on blood pressure in three men and three women with hypertension was measured in an 8-week study. The results showed that diastolic (DBP) and systolic (SBP) blood pressure decreased significantly after 8 weeks of administration of the test composition. Before administration, the SBP was 148.6 mmHg, and after it was 137.0 mmHg. For DBP, before the test it was 95.5 mmHg, and after it was 85.6 mmHg [161].

These results provide a promising basis for further research and potential clinical applications in cardiovascular health. However, rigorous scientific research and clinical trials are needed to fully establish these therapeutic benefits and their underlying mechanisms.

### 10.9. Anti-Nociceptive Activity

To evaluate the potential future therapeutic uses of *C. siliqua*, a study was conducted by Alqudah et al., 2022 [113]. In this study, the ethanol extract of *C. siliqua* leaves was examined using well-established animal models of inflammation and pain. A hot plate latency test (55 °C) was used to assess the analgesic effect of doses of 10, 31.6, 100, and 316 mg/kg of ethanol extracts. In addition, pain was induced in the sub-plantar region of the right hind paw using formalin (5%, 50 μL, purity ≥37%), which was injected 60 min after the treatments. The nociceptive response was recorded by measuring the duration each mouse spent licking the formalin injection site. Licking time was recorded 0–5 min (early phase) and 15–30 min (late phase) after injection. The results of this study demonstrated that the ethanolic extract of *C. siliqua* leaves reduced the paw-licking time in both the early and late phases after formalin injection. The same effect was also observed when the hot plate test was conducted.

### 10.10. Hepatoprotective Activity

A recent study was conducted to investigate the hepatoprotective effect of *C. siliqua*. Microwave-assisted extraction of carob pulp flour was performed; for each experimental run, 10 g of plant material was mixed with 100 mL of respective solvents of water and ethanol. Afterwards, the carob extract (CE) obtained under optimal conditions was analyzed in vivo using a paracetamol-induced hepatotoxicity model in mice. Treatment with CE attenuated the parameters of liver damage, specifically the activity of aspartate and alanine aminotransferases. The extract also prevented the paracetamol-induced increase in malondialdehyde levels. Pretreatment with CE reversed the activities of superoxide dismutase, catalase, glutathione peroxidase, and glutathione S-transferase enzymes after the high dose of paracetamol in the liver. Hepatotoxicity induced using a toxic dose of paracetamol was also observed through histopathological alterations, which were significantly reduced in the groups treated with CE before paracetamol [127].

These findings point to the therapeutic potential of carob extract in the treatment and prevention of liver diseases. However, further investigations and clinical trials are needed to fully elucidate the mechanisms behind these protective effects and to establish safe and effective dosage guidelines for potential therapeutic use.

## 11. Toxicology of *C. siliqua* L.

A study was conducted by Gulay et al., 2012 [162], in which the toxicological properties of *C. siliqua* in male New Zealand white rabbits were studied. In this context, for the treated group, the rabbits received 10 cm^3^ of carob cures by boiling the carob fruit. No toxicological signs or deaths linked to the carob extract were observed during the 7 weeks of the experiment. In the context of this study, with respect to hematological and physiological parameters and histological aspects of organs such as the liver, kidneys, lungs, brain, and heart, the use of carob in animals does not cause toxicological effects and can be used for human consumption.

To determine the acute toxicity of carob, a study was carried out on four groups of female mice. Different doses of the methanolic extract of the bark of *C. siliqua* were administered to the animals. The lethal dose of 50 was greater than 5 g/kg and did not cause mortality or change in general behavior in the mice tested [49]. The hydro-methanolic extract of *C. siliqua* pods was dissolved in normal saline solution and administered at a single dose of 4 g/kg; 8 g/kg; and 12 g/kg. Then, the extract was administered to albino mice (♂/♀) weighing 20 to 30 g. The control group received normal saline only. The general behavior of the mice was continuously monitored for 14 days. Doses of 4 g/kg and 8 g/kg showed no clinical signs of acute toxicity in animals tested. However, mice at a dose of 12 g/kg showed signs of toxicity including increased defecation, decreased mobility, piloerection, and stretching of the forelimbs. No mortality was recorded until the end of the study [159]. Another study of the acute toxicity of *C. siliqua* L. was carried out for 18 Wistar rats divided into three groups: The first group orally received a single dose of carob pod extract at 1500 mg/kg of body weight. The second group received a single oral dose of carob leaf extract at 1500 mg/kg. The third group received distilled water (control group). Animals were observed for the first four hours and periodically for 24 h after treatment. Changes in skin, eyes, mucous membranes, body weight, and behavioral patterns were recorded during the test period (14 days). The results of this study showed that the LD_50_ was higher than 1500 mg/kg, because at this dose, no sign of hypoactivity, morbidity, or lethality was observed in the animals tested [163].

Various studies on the toxicological properties of *C. siliqua* demonstrate a high level of safety for both human and animal consumption, justifying their potential use in the development of therapeutic applications.

Research on carob, while enlightening, has faced certain limitations that need to be addressed for a comprehensive understanding of its potential benefits. While current research has illuminated certain aspects of its properties, there remains a vast reservoir of traditional knowledge spread across continents, waiting to be integrated with modern science. The emphasis on interdisciplinary and collaborative studies not only highlights the progressive direction research should take but also underscores the potential treasures hidden within carob. Only through such a cohesive approach can we ensure that the myriad benefits of carob are explored, understood, and utilized to their fullest potential.

## 12. Conclusions and Perspectives

This review provides a comprehensive overview of *C. siliqua* L., including its taxonomy, cytogeography, and biological significance of the species. From North Africa to Iran, India, and China, carob has been consumed and utilized in various traditional medicinal practices. As a plant of immense value, the carob tree has considerable potential for exploitation in both traditional and contemporary drug development due to its myriad medicinal uses.

The pharmacological studies detailed herein encompass nearly all the ethnobotanical applications of this species, highlighting its potential in a range of pharmacological domains, including antimicrobial, anticancer, antidepressant, and antihyperglycemic activities, among others. A closer look at the phytochemical composition of this plant’s various parts reveals a rich and diverse array of compounds, which potentially underlie its bioactive potential. As such, further phytochemical and pharmacological research is warranted. Such research could focus on bioguided fractionation to isolate and purify potentially biologically active chemical components. It is also crucial to elucidate the mechanisms of action, bioavailability, and pharmacological validation of these active constituents to exploit specific pharmacological activity fully. This research could lay the groundwork for the enhanced utilization of this medicinal plant. Moreover, the development and integration of this plant into clinical practice could prove beneficial. However, prior to this, clinical studies investigating its traditional use are a prerequisite.

## Figures and Tables

**Figure 1 plants-12-03303-f001:**
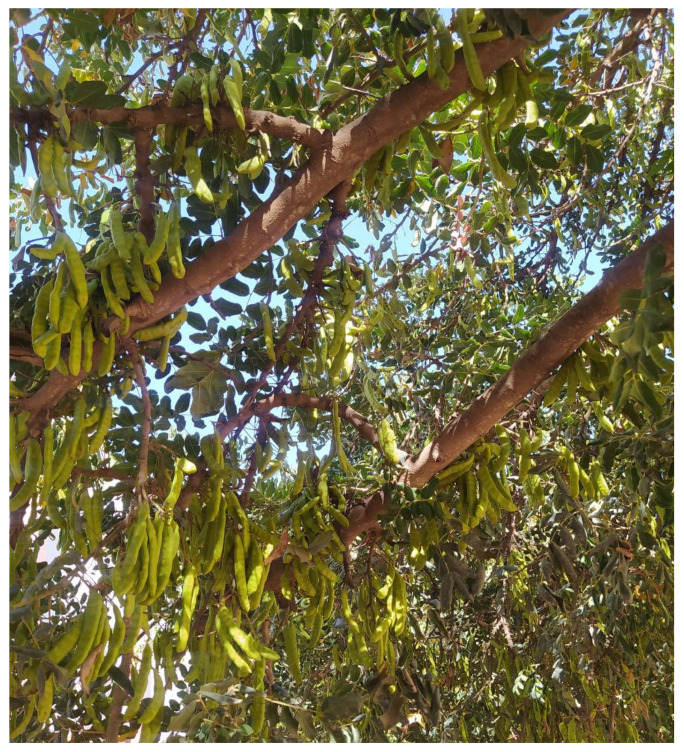
*C. siliqua* L. (photo by W. Dahmani at the Faculty of Sciences, Mohamed First University, Oujda, Morocco).

**Figure 2 plants-12-03303-f002:**
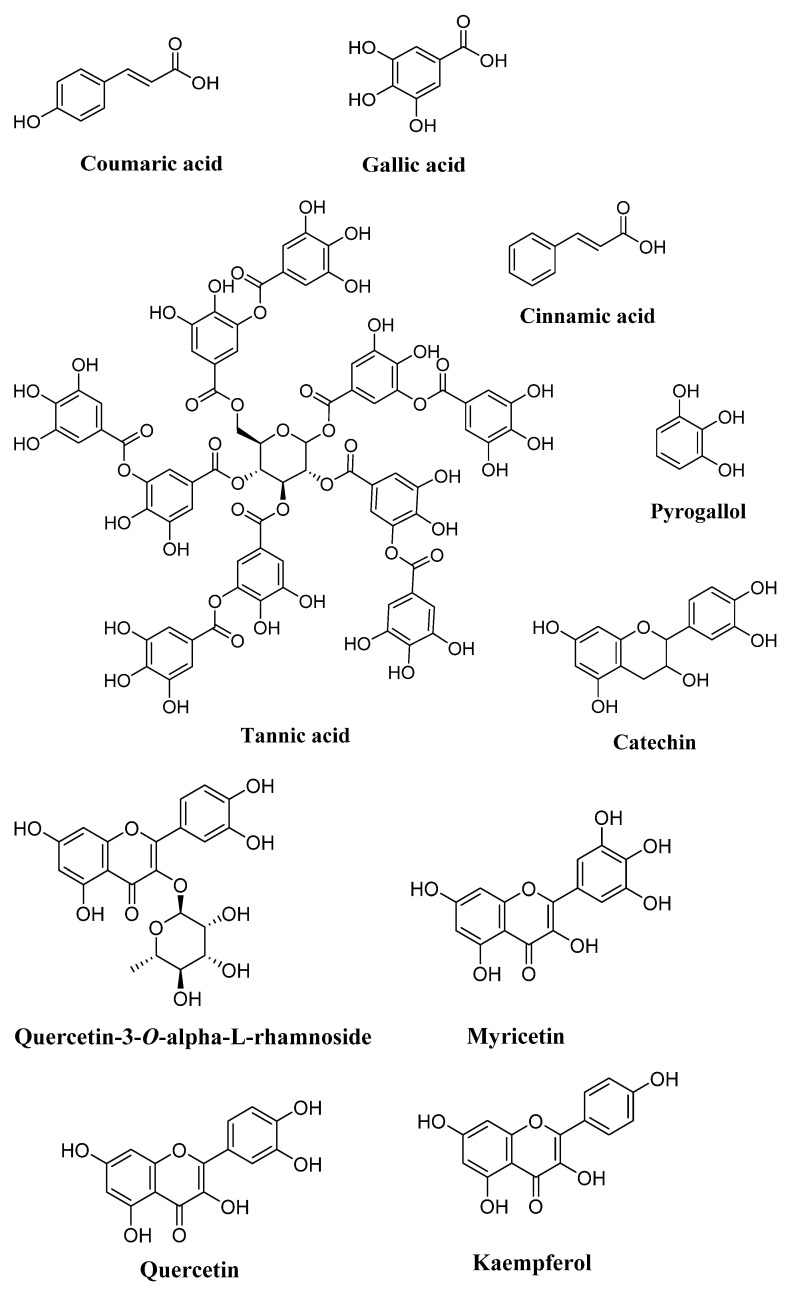
Chemical compounds present in *C. siliqua* L. (drawn with ChemDraw 15.0).

**Table 1 plants-12-03303-t001:** Estimated area cultivated, production, and yield of carob in the Maghreb countries, year 2021 (table based on FAOSTAT data).

Country	Cultivated Area (ha)	Production (Tons)	Yield (t/ha)
Morocco Turkey Lebanon Algeria	10,389 1078 400 688	21,976.85 20,633 4351.1 3219	2.11 19.14 0.01 4.67
Tunisia America Ukraine Jordan	408 80 95 0	818.17 311.93 200.24 0	2.00 3.89 2.10 0
Total	1682,467	138,051,995	32.92

**Table 2 plants-12-03303-t002:** Ethnomedicinal and food uses of carob.

Country	Ethnomedicinal/Food Uses	Part and/or Method of Use	References
Morocco	Diarrhea/toxic for fish	Fruits, barks	[53]
	Intestinal parasites, digestive system, antidiarrheal		
	Digestive, skin, nervous	Leaves, fruits decoction or raw	[54]
	Diabetes	Seeds, leaves or fruits infusion, powder, or decoction	[51,52]
	Digestive diseases	Leaves, seeds decoction or powder	[51]
		Fruits powder	[55]
Tunisia	Food	Pulp	[25]
	Gastrointestinal disorders, Diarrhea	Fruits	[50]
Egypt	Diarrhea	Fruits, leaves infusion	[56]
Jordan	Diabetes	Leaves decoction cold or hot beverage of the pods	[57]
	Cough		[58]
Lebanon	Sweetener	Molasse of ripened fruits	[59]
	Food	Pods	
	Medicine	*Not indicated*	[60]
Palestine	Food	Raw fruits	[61]
	Hypertension	Raw or cooked fruits, seeds	[7]
Iraq	Abdominal pain, diarrhea	Fruits	[62]
Turkey	Diuretic, purgative	Fresh fruits	[63]
	Diarrhea	Leaves	[44]
Cyprus	Diabetes	Fruits decoction	[64]
	Laxative, demulcent	Pods	[65]
Greco-Arab	Diabetes, herpes, lip sores	Leaves decoction	[66]
India	Hypocholesterolemic, hypolipidemic, hypoglycemic, demulcent, resolvent	Not indicated	[67]
Iran	Menorrhagia	Patient should sit in a pot of a decoction of several plants	[68]
Spain	Chocolate or coffee substitute, preservative for olives	Fruits, leaves	[16]
Sicily	Food	Raw or boiled fruits	[69]
Italy	Emollient	Decoction with *Ficus carica* L. and *Malva sylvestris* L.	[70]

**Table 3 plants-12-03303-t003:** Chemical composition of carob (*C. siliqua* L.).

Class	Compounds	Part of the Plant	References
Phenols	Resorcinol	Leaves, pods, pulp, seeds	[102]
	Vanillin, fraxidin, 2,4- bis(dimethylbenzyl)-6-butylphenol	Leaves	[102,109]
	Alizarin, hydroquinone, lignan bis(trihydroxyphenyl)methanone	Pods	[110,111]
Phenolic acids	Gallic acid, chlorogenic acid, syringic acid, ferulic acid, coumaric acid, cinnamic acid	Leaves, pods, pulp, seeds	[4,5,89,102,110,111,112,113,114,115,116,117,118,119,120,121,122,123,124,125,126,127,128,129,130]
	4-Hydroxybenzoic acid, caffeic acid, vanillic acid, gentisic acid	Leaves, pods, pulp	[5,89,102,112,113,114,119,121,122,125,126,127,128,129]
	Tannic acid	Leaves, pods, seeds	[4,122]
	Ellagic acid, rosmarinic acid	Pods, pulp, seeds	[111,119,125,126,127]
	Sinapic acid	Pulp, seeds	[111,126]
	Pyrogallol, methyl gallate, benzoic acid, protocatechuic acid	Pods, pulp	[4,5,110,111,119,120,122,128,129]
	Quinic acid	Leaves, pods	[109,120]
	Transferulic acid, *O*-feruloylrutinose, *O*- feruloylrutinose isomer, *p*-coumaroyl- galloylhexose, *O*-*p*-coumaroylrutinose, siliquapyranone	Pods	[110,120,125]
	4-Hydroxy-coumaric acid	Leaves	[112,113]
	5-Caffeoylquinic acid, myristic acid, ascorbic acid	Pulp	[111,126,131]
Flavonoids	Epicatechin, quercetin, kaempferol, luteolin, catechin, apigenin	Leaves, pods, pulp, seeds	[4,5,89,102,110,111,112,113,115,116,118,119,120,121,122,123,124,125,126,127,128,130,131,132]
	Epigallocatechin gallate, rutin, myricetin, naringenin	Leaves, pods, pulp	[4,5,110,111,112,113,114,116,117,118,119,120,121,123,124,125,126,127,128]
	Iso-rhamnetin	Leaves, pods, seeds	[102,110,111,119,125]
	Leucoanthocyanins	Leaves, pulp, seeds	[133]
	Genistein	Leaves, pods	[5,119]
	Quercitrin, catechin tannins	Leaves, pulp	[112,113,115,127,133]
	Anthocyanins	Pods, pulp, seeds	[5,134,135]
	Myricitrin, daidzein, flavonol, morin	Leaves	[102,112,113,114,116]
	Rhamnosides, chrysoeriol, tricetin dimethyl ether, (iso)schaftoside-4′-*O*- glucoside, gallocatechin, chrysoeriol-*O*- deoxyheoxoside, dihydroxyflavanone hexoside, tetrahydroxy flavanone, trihy- droxy flavone (apigenin isomer), kamp- feride, methoxykampferol, dihydroxy flavanone, tricetin dimethyl ether, cirsi- liol, flavone glycosides, hydroxytyrosol	Pods	[5,110,119,120,122]
	Crismaritin, catechol, isoquercetrin, flavonols 3′,4′,5,7-OH, 2-hexadecanol scutellarin tetramethyl ether, silybin B, hydroxytyrosol, catechin gallate	Pulp	[4,126,129,131]
	Apigenin flavone, chrysin aglycones	Seeds	[119]

## Data Availability

Not applicable.

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
