# Peer review of "Exploring Carob (Ceratonia siliqua L.): A Comprehensive Assessment of Its Characteristics, Ethnomedicinal Uses, Phytochemical Aspects, and Pharmacological Activities"

_plants, 2023, doi:10.3390/plants12183303_

Round 1

Reviewer 1 Report

It is a well-documented manuscript covering many aspects of Ceratonia siliqua L.

I have only some minor comments and/or questions on some points:

lines 297: endo-sperm; 301: com-mercial; 310: choco-late; 312: theo-bromine

I think that these words were involuntarilly split during the transfer.

line 370: "quercetin-3-O-glucoside" should be corrected as "quercetin-3-O-glucoside". Similar cases are also present in Table 3.

line 415: ref. Özcan et al. should be given as Özcan, M.M. in the line 1050.

Questions:

lines 451 - 455: This statement needs some details. What are the five macro- and thirty microelements in the whole pod, pulp, and seed? 

Author Response

Dear reviewer,

We thank you very much for your kind comments. We greatly appreciate your thoughtful comments. We have taken all your remarks into consideration:

lines 297: endo-sperm; 301: com-mercial; 310: choco-late; 312: theo-bromine:

Response: The software “Word” separates the words which are at the end of the lines automatically, however we have corrected the separated words which were in the middle of the text.

line 370: "quercetin-3-O-glucoside" should be corrected as "quercetin-3-O-glucoside". Similar cases are also present in Table 3.

Response: It was corrected in the text and in the table.

line 415: ref. Özcan et al. should be given as Özcan, M.M. in the line 1050.

Response: this reference was corrected in the list of references.

lines 451 - 455: This statement needs some details. What are the five macro- and thirty microelements in the whole pod, pulp, and seed? 

Response: Some details about these elements have been added to the text.

Reviewer 2 Report

1- The Abstract is too short and it should be revised, and more information on chemical components and pharmacological benefits of carob tree presents in the manuscript.

2- Keywords should be written on the basis of Alphabetic order, and it is suggested to do not use words which have been used in Title of the manuscript.

3- Please, do not use scientific word as the key word in your article (Ceratonia siliqua), and the keywords should be between 5-7.

4- Paragraphing in Introduction part is not clear at all. Each paragraph should start with new contents, and this way of writing and paragraphing should be corrected.

5- At the first line, authors should first write English name for example Caron and then scientific name in paranthesis.

6- Please, write and add Leguminosae after Fabaceae at the first line of Introduction. 

7-Line 45 and 46, Rtibi et al. (2018) (9)... 2018????

8- In section 2, when authors write different names of Carob tree in Hebrew, Arabic, English and French, they should write it in other different languages such as Persian, Chinese, Russian, Italians, Spanish, etc.

9- In line 66, authors have written Leguminosae after Fabaceae, and it has been mentioned, it should be done at the first part of article, and one time of mentioning is enough for it.

10- Paragraphing and managing sentences in all parts of the manuscript has many problems, which need to be revised completely.

11- Why do authors in some parts of the article have written Ceratonia siliqua L., and in other parts C. siliqua L.??? if one time authors write Ceratonia siliqua L., they can write C. siliqua after that in other parts of the manuscript.

12- Line 160, Mitrakos (1988)???? Do authors check the format of manuscript?

13- Line 198, Correia et al. (2010) (42)????

14- Authors need to design one table and mentioned all pharmacological and health benefits of the plant in one table with References.

15- Most parts of the article is like Copy and Paste, even in one Review article authors should give their own ideas and opinions as well.

16- Conclusion needs to be written and revised completely, this sentence is very meaningless and not appropriate for the Conclusion parts:

The purpose of this review was ...... in the future (Line 796-797-798).

17- The articles needs one part after conclusion which is called Abbreviation part.

18- References are not on the basis of journal s format... check reference 1 (researh J of Phar and Technology); Reference 5, Reference 6, Reference and all others.

19- All articles needs DOI and authors should add their DOI in the reference list.

The English language should be revised and double-checked by the authors and one native English speaker. There are many mistakes and errors in the current format.

Author Response

Dear reviewer,

We greatly appreciate your thoughtful comments. We have taken all your remarks into consideration:

1- The Abstract is too short and it should be revised, and more information on chemical components and pharmacological benefits of carob tree presents in the manuscript.

Response: The abstract was completed by more information about phytochemical components and pharmacological properties of the plant.

2- Keywords should be written on the basis of Alphabetic order, and it is suggested to do not use words which have been used in Title of the manuscript.

Response: Keywords are in alphabetical order, and we removed the words that were already in the title.

3- Please, do not use scientific word as the key word in your article (Ceratonia siliqua), and the keywords should be between 5-7.

Response: The scientific name of the plant has been removed from the keywords.

4- Paragraphing in Introduction part is not clear at all. Each paragraph should start with new contents, and this way of writing and paragraphing should be corrected.

Response: We reworded the introduction to make it clearer.

5- At the first line, authors should first write English name for example Carob and then scientific name in paranthesis.

Response: the correction has been done in the first line of the text.

6- Please, write and add Leguminosae after Fabaceae at the first line of Introduction. 

Response: The term Leguminosae has been added to the first line of the introduction.

7-Line 45 and 46, Rtibi et al. (2018) (9)... 2018????

13- Line 198, Correia et al. (2010) (42)????

Response: It was corrected for all the citations.

8- In section 2, when authors write different names of Carob tree in Hebrew, Arabic, English and French, they should write it in other different languages such as Persian, Chinese, Russian, Italians, Spanish, etc.

Response: Some other names of carob in different languages have been added to the text.

9- In line 66, authors have written Leguminosae after Fabaceae, and it has been mentioned, it should be done at the first part of article, and one time of mentioning is enough for it.

Response: It was corrected, we removed Leguminosae from line 66.

10- Paragraphing and managing sentences in all parts of the manuscript has many problems, which need to be revised completely.

Response: We have tried to revise all parts of the manuscript to correct errors, and paragraphing.

11- Why do authors in some parts of the article have written Ceratonia siliqua L., and in other parts C. siliqua L.??? if one time authors write Ceratonia siliqua L., they can write C. siliqua after that in other parts of the manuscript.

Response: It was changed throughout the manuscript by C. siliqua.

12- Line 160, Mitrakos (1988)???? Do authors check the format of manuscript?

Response: The correction has been done.

14- Authors need to design one table and mentioned all pharmacological and health benefits of the plant in one table with References.

Response: We already have 3 tables: the first specifies the cultivated area, production, and yield of carob in some countries, the second includes its ethnomedicinal and food uses, and a third contains its chemical composition. Concerning pharmacology it was very detailed in the text by specifying various activity and we considered that it was preferable to present it in the text. Another additional table will take a lot of time, make the article heavier and cannot contain all information given in the text.

15- Most parts of the article is like Copy and Paste, even in one Review article authors should give their own ideas and opinions as well.

Response: Regarding this point, we have made a lot of efforts to avoid any kind of plagiarism. However, since it is a review article that brings together the ideas of other articles, it is possible to find certain passages of the articles cited in the references. Nevertheless, the rate of similarity concerning our article is very low.

16- Conclusion needs to be written and revised completely, this sentence is very meaningless and not appropriate for the Conclusion parts:

The purpose of this review was ...... in the future (Line 796-797-798).

Response: We modified the conclusion, and we deleted this sentence from it.

17- The articles needs one part after conclusion which is called Abbreviation part.

Response: An abbreviation list was added after the conclusion.

18- References are not on the basis of journal s format... check reference 1 (researh J of Phar and Technology); Reference 5, Reference 6, Reference and all others.

Response: We worked the references automatically using endnote, and we chose the MDPI log format. We found it very difficult to work manually afterwards.

19- All articles needs DOI and authors should add their DOI in the reference list.

Response: The list of references has been adjusted following the instructions for authors, and DOIs have not been requested to be added.

Comments on the Quality of English Language

The English language should be revised and double-checked by the authors and one native English speaker. There are many mistakes and errors in the current format.

Response: The English of the manuscript has been revised by one English teacher in our university and double-checked by the authors.

Reviewer 3 Report

This review concerning the characterization, ethnomedicinal, phytochemistry and pharmacology of carobs is acceptable with major revision.

1.  Please add some important references but not list in the present MS to update this topic.  Food Sci & Nutrition, 2023, 11, 3641-3654 (Nutritional, biochemical, and clinical applications of carob: A review); Food Chemistry, 2018, 269, 355-374 (Polyphenols in carobs: A review on their composition, antioxidant capacity and cytotoxic effects, and health impact); Foods, 2022, 11, 2154 (Carob: A Sustainable Opportunity for Metabolic Health)

2. Please follow the author guide to unify the reference format.

Author Response

Dear reviewer,

We greatly appreciate your thoughtful comments. We have taken all your remarks into consideration:

  1. Please add some important references but not list in the present MS to update this topic.  Food Sci & Nutrition, 2023, 11, 3641-3654 (Nutritional, biochemical, and clinical applications of carob: A review); Food Chemistry, 2018, 269, 355-374 (Polyphenols in carobs: A review on their composition, antioxidant capacity and cytotoxic effects, and health impact); Foods, 2022, 11, 2154 (Carob: A Sustainable Opportunity for Metabolic Health)

Response: As suggested by the reviewer 3, the following references have been added to the manuscript:

  1. Gioxari, A.; Amerikanou, C.; Nestoridi, I.; Gourgari, E.; Pratsinis, H.; Kalogeropoulos, N.; Andrikopoulos, N.K.; Kaliora, A.C. Carob: A sustainable opportunity for metabolic health. Foods 2022, 11, 2154. https://doi.org/10.3390/foods11142154

  1. 144. Stavrou, I.J.; Christou, A.; Kapnissi-Christodoulou, C.P. Polyphenols in carobs: A review on their composition, antioxidant capacity and cytotoxic effects, and health impact. Food chemistry 2018, 269, 355-374. https://doi.org/10.1016/j.foodchem.2018.06.152

  1. Ikram, A.; Waseem, K.; Wajeeha Zafar, K.u.; Ali, A.; Afzal, M.F.; Aziz, A.; Faiz ul Rasool, I.; Al‐Farga, A.; Aqlan, F.; Koraqi, H. Nutritional, biochemical, and clinical applications of carob: A review. Food Science & Nutrition 2023, 11, 3641–3654. https://doi.org/10.1002/fsn3.3367

  1. Please follow the author guide to unify the reference format.

Response: The references were added with a bibliography software (EndNote) where the reference format was chosen following the style of the MDPI journal.

Round 2

Reviewer 2 Report

There are Some revisions which should be considered, and the manuscript should be checked after revision.

1) Delete Review in Title.

2) Keywords should be different from words in the manuscript.

3) Keywords should be written on the basis of alphabetic format and all should start with small word.

4) The manuscript needs one section (( Materials and Methods )) which should be written after Introduction.

5) It is suggested to design one table, like Table 1, and in that table, estimated area of cultivation of Carob mentioned in other parts of the world, for example in China, India, the USA, and other countries in the Middle East and North Africa.

6) For title (7), use another word instead of USE, the authors can use, APPLICATIONS OF CAROB.

7) There are too many paragraphs in the whole manuscript, and it should be revised, and in the new paragraphing system, all paragraphs start with new information.

8) 7.1; it is better to use INDUSTRIAL APPLICATIONS than Industrial uses.

9) Line 283, Vladimir et al. (2018). Why do not authors consider format of MDPI journals?????

10) Line 328, A study carried out by Fadel et al. in 2011... it should be corrected.

11) Line 461, El Bouzdoui et al. (2017). It is not on the basis of the format of journal.

12) Line 471, Khalifa et al. (2013), it should be corrected.

13) Line 479, according to Fidan et al. (2020), it should be corrected.

14) Line 485, Dallali et al. (2018), it should be corrected.

15) Line 497, it should be corrected.

16) Line 518, it should be corrected.

17) Line 521 and 527, references should be written on the basis of journal s format.

18) Line 621, the reference should written on the basis of journal s format.

19) All references in the text should be corrected.

20) For Abbreviation part, authors do not use to design a table, like Table 12, the abbreviation list should be designed after conclusion, before references, but not in the Table.

21) All references need DOI.

22) All references should be written on the basis of journal s format, they can not be accepted in the current format.

23) Please check instructions for authors for all parts of the manuscript, especially writing References.

The article needs Minor English revision.

Author Response

Dear Reviewer,

Thank you for your thorough review of our article. We have carefully considered and addressed your comments.

Best regards,

Reviewer 3 Report

Acceptable in the present format

Author Response

Dear Reviewer,

Thank you very much for your response.

Best regards,

Round 3

Reviewer 2 Report

The article has been revised, and now it has improved and the revised part has been shown in RED color.

However, authors should delete YEARS in the manuscript, and corrected References accordingly.

Authors have also used many paragraphs, in many cases in this manuscript, authors can join paragraphs, and paragraphs can be joined together, for example 2 or 3 paragraphs can be changed into ONE paragraph.

I also recommend authors write a paragraph before conclusion and mention some limitation about researches about carob and the future direction which should be considered for future researches, it is also important and suggested to make a link between traditional medicinal sciences, where people use and consume Carob, for example from North Africa to Iran, India, China, etc. Do authors check the application of this important medicinal plants in different traditional Asian medicinal sciences???

But all in all, the article has written very well, and the topic is interesting, and it just need Minor revision.

Author Response

Reviewer 2 Round 3

The article has been revised, and now it has improved and the revised part has been shown in RED color.

However, authors should delete YEARS in the manuscript, and corrected References accordingly.

Response: We formatted the citations in our manuscript based on the style used in recent articles from Plants, such as the one by Malmir et al., 2022 (Plants 2022, 11(22), 3173; https://doi.org/10.3390/plants11223173)

Authors have also used many paragraphs, in many cases in this manuscript, authors can join paragraphs, and paragraphs can be joined together, for example 2 or 3 paragraphs can be changed into ONE paragraph.

Response: We further reduced the paragraphs by joining some of them.

I also recommend authors write a paragraph before conclusion and mention some limitation about researches about carob and the future direction which should be considered for future researches, it is also important and suggested to make a link between traditional medicinal sciences, where people use and consume Carob, for example from North Africa to Iran, India, China, etc. Do authors check the application of this important medicinal plants in different traditional Asian medicinal sciences???

Response: We added a paragraph before the conclusion and incorporated a sentence into conclusion in RED color based on your feedback.

But all in all, the article has written very well, and the topic is interesting, and it just need Minor revision.

Response: Thank you for your valuable comments, which helped us improve the manuscript.
